



# Estimation of local training data point densities to support the assessment of spatial prediction uncertainty

Fabian Lukas Schumacher[1,2], Christian Knoth[2], Marvin Ludwig[1], and Hanna Meyer[1]

[1]University of Münster, Institute of Landscape Ecology, Heisenbergstr. 2, 48149 Münster, Germany
[2]University of Münster, Institute for Geoinformatics, Heisenbergstr. 2, 48149 Münster, Germany

**Correspondence:** Fabian Lukas Schumacher (fabian.schumacher@uni-muenster.de)

**Abstract.** Machine learning is frequently used in environmental and earth sciences to produce spatial or spatio-temporal predictions of environmental variables based on limited field samples - increasingly even on a global scale and far beyond the location of available training data. Since new geographic space often goes along with new environmental properties represented by the model's predictors, and since machine learning models do not perform well in extrapolation, this raises questions
regarding the applicability of the trained models at the prediction locations.

Methods to assess the area of applicability of spatial prediction models have been recently suggested and applied. These are typically based on distances in the predictor space between the prediction data and the nearest reference data point to represent the similarity to the training data. However, we assume that the density of the training data in the predictor space, i.e. how well an environment is represented in a model, is highly decisive for the prediction quality and complements the consideration of
distances.

We therefore suggest a local training data point density (LPD) approach. The LPD is a quantitative measure that indicates, for a new prediction location, how many similar reference data points have been included in the model training. Similarity here is defined by the dissimilarity threshold introduced by Meyer and Pebesma (2021) which is the maximum distance to a nearest training data point in the predictor space as observed during cross-validation. We assess the suitability of the approach in a
simulation study and illustrate how the method can be used in real-world applications.

The simulation study indicated a positive relationship between LPD and prediction performance and highlights the value of the approach compared to the consideration of the distance to a nearest data point only. We therefore suggest the calculation of the LPD to support the assessment of prediction uncertainties.

## 1   Introduction

Machine learning methods play an important role in the spatial modelling and prediction of environmental variables especially when relationships are assumed to be highly complex and non-linear. In a typical spatial prediction workflow in environmental science and ecology, machine learning algorithms are employed to train models using reference data gathered from local field observations and gridded predictors, which are often derived from remote sensing data. The resulting model is then deployed to make predictions beyond the training locations, i.e. to map the variable of interest. In this way, a large number of



environmental variables have been mapped, such as soil nematode abundance (van den Hoogen et al., 2019), soil bacteria (Delgado-Baquerizo et al., 2018), soil properties (Hengl et al., 2017) or global tree restoration potential (Bastin et al., 2019) to mention just a few global applications. While machine learning excels at predicting nonlinear relationships, its effectiveness diminishes when transferring models to new environments divergent from the training data, due to its limited capabilities in extrapolation (Ludwig et al., 2023). Field data, however, are often sparse and not evenly distributed over the area of interest,

hence spatial prediction frequently requires predictions to "unknown environments" (Meyer and Pebesma, 2021). This is especially obvious in global mapping approaches where, e.g., biodiversity surveys, soil sampling campaigns or climate stations are often strongly clustered in Europe or Central America while other geographic areas (and hence likely other environments) are poorly covered by data. Despite the recognition of this limitation, the lack of data in certain environments is yet rarely considered in the presentation and discussion of machine learning-based prediction maps of the environment. Meyer and Pebesma

(2022) therefore initiated a discussion emphasizing the necessity to improve the assessment and communication of limitations in spatial predictions.

    To assess these limitations, Meyer and Pebesma (2021) recently suggested a method to derive the "area of applicability" (AOA) of prediction models. The AOA is based on the dissimilarity (in terms of predictor values) of each new prediction location to the training data by using the normalized distance to the nearest training data point in the weighted multidimensional

predictor space, referred to as the dissimilarity index (DI). The binary AOA is derived by applying a threshold to the DI. Other approaches to limit predictions to the area where the model has been enabled to learn about relationships are the extrapolation index (Jung et al., 2020), convex hulls in the feature space (van den Hoogen et al., 2021) or the use of geographic distances from training data locations (Sabatini et al., 2022). All these approaches share the limitation that they do not discriminate between areas with few or even just solitary training data points, and areas that are densely covered by training data.

Figure 1 illustrates the problem of purely distance-based approaches, here using the DI suggested by Meyer and Pebesma (2021). In figure 1a, both prediction data points have the same DI (i.e. same distance to a nearest training data point), although predictions made for point $A$ are based on a large number of similar training data while the prediction for data point $B$ is not well supported by training data. Figure 1b shows a hypothetical classification example, that emphasizes the relevance of accounting for data point densities rather than relying solely on the distance to a nearest training point. We assume that predictions for

data points falling in regions in the predictor space that are densely covered by training data (point $A$ in figure 1a) exhibit lower prediction uncertainty compared to predictions that rely on sparse and isolated training data points (point $B$ in figure 1a).

    We therefore suggest that the training data point densities are highly informative to assess the prediction uncertainties. However, common approaches to estimate the density, such as Kernel Density Estimation, are hardly feasible in the multidimensional predictor data space, as with increasing dimensionality the computational and memory effort increases drastically.

As an alternative, we suggest calculating local training data point densities, which indicate the density of training data in a defined neighborhood around a new prediction location. We build on the suggested AOA threshold provided by Meyer and Pebesma (2021) to define similarity and count the absolute number of similar training data points within the given distance in the predictor space and call it the local training data point density (LPD). When calculated for each prediction location, the method allows for mapping the LPD.




Here we describe the method to calculate the LPD and show its use in a simulation study as well as in a real-world case study. We test the suitability of the LPD to reflect the prediction error in comparison to the DI as a nearest neighbor distance-based index.

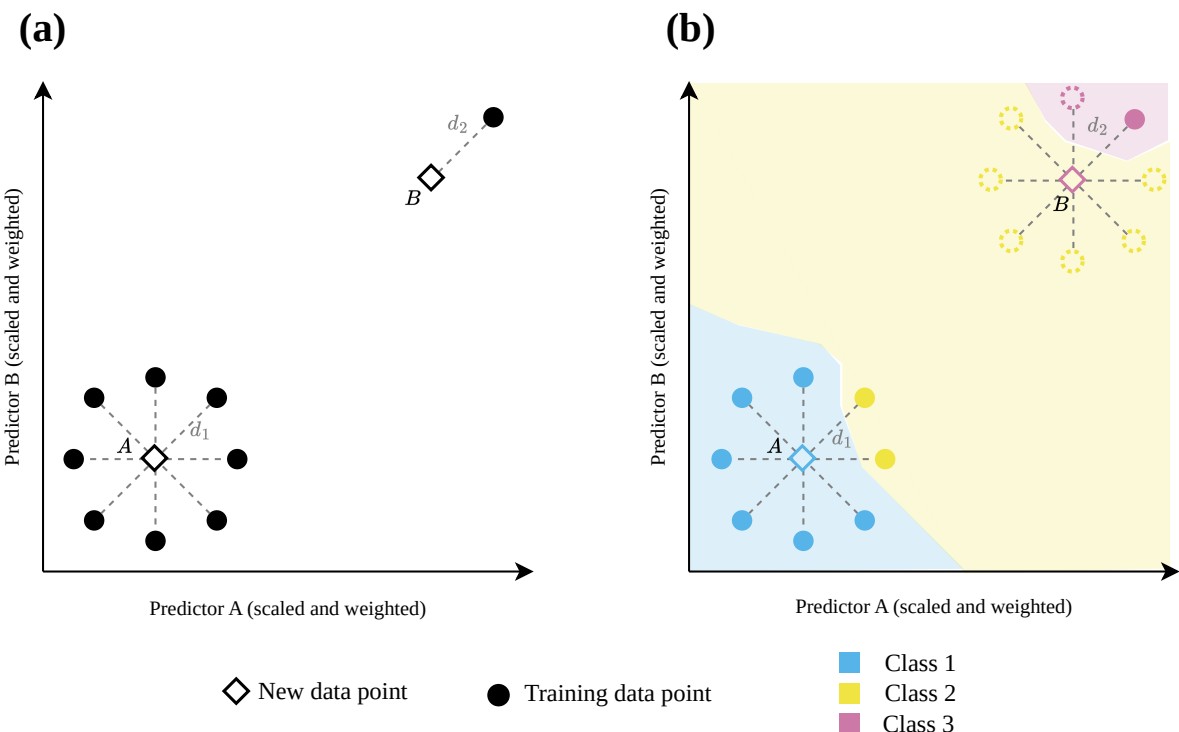

**Figure 1.** Hypothetical 2-dimensional predictor data space with training data points (circle) and new prediction data points (diamond) to illustrate the motivation of this study. In (a) both new prediction data points have the same distance ($d_1 = d_2$) to their nearest neighbor in the training data set, and consequently they have the same dissimilarity index DI ($DI_1 = DI_2$). (b) shows at the example of a classification task, why the distance to the nearest training data point alone may not be a good indicator for prediction uncertainty. The new data point $B$ would, according to most classification algorithms, be classified as *Class 3*, whereas only the consideration of multiple training points (dotted circles) may reveal the appropriate assignment to *Class 2*. Even though the new prediction data point is inside the area of applicability (AOA) because of a low DI, we would consider the prediction uncertainty as being high, due to the low support of training data (N=1 compared to N=8 in case A).



## 2   Materials and Methods

We suggest a quantitative measure that indicates, for a new data point (i.e., a prediction location), how many similar reference
data points have been included in the model training. In a simulation study as well as a real-world application, we show how
the method can be used and test the ability of the method to reflect prediction uncertainties.

### 2.1   Calculation of the local data point density

We define similarity by the DI threshold introduced by Meyer and Pebesma (2021) to derive the AOA, which is the maximum
distance to a nearest training data point in the predictor space (without outliers) as observed during model cross-validation.
The threshold is calculated as follows (details in Meyer and Pebesma, 2021):

- Standardization of predictor variables: to ensure that all variables are treated equally, the predictor variables are scaled
  by dividing mean-centered values by their respective standard deviations.

- Weighting of variables: scaled variables are multiplied with their non-standardized importance estimates to account for
  the different relevance within the model.

- Dissimilarity index calculation under consideration of cross-validation: per default, the euclidean distance is calculated
  for each training data point and its nearest neighbor **not** located in the same cross-validation fold. The distances are then
  divided by the average of the distances in the training data.

- Delineation of the threshold: per default, the outlier-removed maximum DI of all training data is used as threshold for
  the AOA where outliers are defined as values greater than the upper whisker (i.e. larger than the 75-percentile plus 1.5
  times the IQR of the DI values of the cross-validated training data).

The suggested local data point density (LPD) is an absolute count of training data points for which this threshold is not
exceeded. Figure 2 shows three examples: the similarity range is per definition identical for all three prediction locations
(diamond). The prediction location *A* has an LPD of 10, since 10 training data points are within the similarity range. For
prediction location *B*, only one training data point is considered similar. Prediction location *C* is outside the AOA and, as a
consequence, the LPD is 0, because no training point is considered similar.

Technically, the LPD is derived by calculating the distance, in the predictor space, between each prediction data point and
each training data point. The similarity threshold is then applied to identify, for any prediction data point (i.e., at any prediction
location), the number of training data points that meet the similarity condition. Additionally, the LPD is also calculated within
the training data set itself, analog to Meyer and Pebesma (2021) for the DI of the training data, by taking the cross-validation
folds into account. This is done to allow for an analysis of the relationship between LPD and the model performance (prediction
accuracy).



## 2.2 Relationship between LPD and prediction accuracy

Meyer and Pebesma (2021) proposed to use the DI to estimate prediction uncertainty based on the DI and the corresponding cross-validation performance. The assumption is that the prediction accuracy decreases with increasing dissimilarity. We suggest analogously, to assess the relationship between LPD and cross-validation performance, assuming that the prediction accuracy decreases with decreasing LPD, due to low support by training data. We assume that this provides a more comprehensive estimate of the model performance. Just as in the implementation of Meyer and Pebesma (2021), we suggest a sliding window across the LPD values to estimate performance metrics derived from the cross-validated training data. The results are used to fit a parametric model to predict the expected performance from the LPD values.

## 2.3 Simulation study

To test the suitability of the LPD approach to reflect prediction uncertainties caused by insufficient training data, we simulated a prediction scenarios where the true response values was known. We used the R package `virtualspecies` associated with the simulation approach of Leroy et al. (2016), which was developed to generate virtual species distributions. Comparable to Meyer and Pebesma (2021), we used the approach to generate a virtual response variable across Europe based on bioclimatic predictor variables (www.worldclim.org/bioclim). Specifically, we downloaded the raster data of 19 bioclimatic variables from the WorldClim dataset (Fick and Hijmans, 2017) at a spatial resolution of 10 minutes and cropped the individual grids to the area of Europe. A subset of these variables was then used to simulate the response variable. Analogous to the simulation study by Meyer and Pebesma (2021), we used the following variables: mean diurnal range ('bio2'), maximum temperature of the warmest month ('bio5'), mean temperature of the warmest quarter ('bio10'), precipitation of the wettest month ('bio13'), precipitation of the driest month ('bio14'), and precipitation of the coldest quarter ('bio19').

The response variable was then simulated based on these variables using the principle component analysis approach described in Leroy et al. (2016). The response to the first two principal components was determined and combined to formulate the final response variable. In alignment with Leroy et al. (2016), Gaussian functions were used to determine the response to the individual axes of the principle component analysis. Therefore, the means of the Gaussian response functions corresponding to these axes were set to 3 for the first axis and $-1$ for the second axis. The standard deviation was set to 2 for both axes. The simulated response variable is shown in Figure 3.

To account for different sampling designs, we simulated two scenarios of different sampling locations each with $N = 100$ locations. The scenarios differed in the spatial distributions of the simulated reference locations: *random* and *clustered* (see 3b). Uniform random sampling was used to draw the *random* samples. The *clustered* samples were drawn in a two-step sampling process which randomly draws $n_{parent}$ samples, before drawing another $\frac{n - n_{parent}}{n}$ points in the buffer radius of each parent sample. 10 parents with a buffer of 100 km were chosen, resulting in 10 clusters containing 10 sample points each.

For each sampling scenario, we created a training data set by extracting the 19 predictor variables and the response. Next, we fitted a random forest model for both scenarios. Random forest was chosen here as it is one of the most commonly applied machine learning algorithms in the context of spatial mapping. We used the random forest implementation by Liaw and Wiener



(2002) accessed via the `caret` R package (Kuhn and Max, 2008). The number of trees was set to 500 in each model and the tuning parameter mtry (number of randomly selected predictors at each split) was tuned between 2 and 19 (number of predictor variables). The cross-validation strategies were designed in line with the distribution of the sampling locations: for both scenarios a 10-fold Nearest Neighbor Distance Matching Cross-Validation by Linnenbrink et al. (2024) was applied (note that for the *random* scenario, this appropriately resulted in a random CV). Analogous to the underlying NNDM LOO

CV by Milà et al. (2022), this method tries to create a k-fold cross-validation design for which the distribution of the nearest neighbor distances (NNDs) between testing and training locations approximately matches the distribution of the NNDs between prediction and training locations. Hence, it produces prediction situations during cross-validation (in terms of geographic distances) that are comparable to those met when predicting the final map based on the trained model.

Both models were then used to make predictions for the whole prediction area (Europe; 82405 pixels), i.e. to map the virtual

response variable. Since the response variable has been simulated, we could derive the true map accuracy, which we describe here as the true absolute error of the predictions. We then calculated the DI and the LPD of the predictions and we assessed the AOA of each of the two models. The variables were ranked according to their importance in the model (Figure A3. To analyze the informative value of the LPD, we compared the LPD values with the true absolute error values of each prediction location. The AOA indicates the area where we assume that, on average, the cross-validation performance applies. Variability in the

prediction performance within the AOA call for a more detailed assessment of the performance using a quantitative measure, such as the DI or the suggested LPD. Therefore, we compared the true root mean squared error(RMSE; mean difference between predicted and true values) for all prediction locations along the range of LPD values (as well as DI values), to the cross-validation estimate of the model.

Since in real-world applications, the comparison between the LPD and the true prediction error is not possible due to the lack

of the spatially continuous reference, we further show how the cross-validation can be used to assess the relationship between LPD (as well as DI) and the prediction performance. We used the models from both sampling designs to exemplify this approach. In moving windows of the LPD we calculated the RMSE of the predictions and assessed the relationship via shape constrained additive models (according to Meyer and Pebesma, 2021), to analyze if the LPD informs about the uncertainty of the predictions. Analogous, we additionally used the DI as an estimator of prediction accuracy (RMSE) and and calculated the

absolute residuals to the true values to see if the LPD can further explain the uncertainty caused by the DI as an estimator of the model performance.

## 2.4 Case study

To demonstrate the applicability of the LPD approach with real-world data, we used an example of mapping plant species richness for the entire region of South America (see Meyer et al., 2024a, for more details on the case study). Again, the

19 bioclimatic variables from the WorldClim dataset were used as predictor variables. Elevation was used as an additional variable, here assuming that bioclimate and elevation are drivers of plant species richness in South America. 703 training locations including the outcome variable (reference data) were derived from the sPlotOpen database described in Sabatini et al.





(2021) and as shown in Figure 4b (datasets used: Pauchard et al., 2013; Peyre et al., 2015; Lopez-Gonzalez et al., 2011; Vibrans et al., 2020).

We fit a random forest model with spatial forward feature selection as described in Meyer et al. (2018) using 5-fold Nearest Neighbor Distance Matching cross-validation for model tuning and variable selection. The identified predictor variables were annual mean temperature ('bio1'), mean temperature of the wettest quarter ('bio8'), annual precipitation ('bio12'), precipitation of driest month ('bio14'), and elevation ('elev') (see Figure 4a). The trained random forest model was used to predict plant species richness for the whole prediction data set (226962 pixels), i.e. the whole area of South America. Subsequently, the

DI and LPD were assessed and the AOA of the prediction model was derived. The variables were ranked according to their importance in the model (Figure B1. We analyzed the relationship between LPD, DI and the prediction error based on the cross-validation. The model was used to make spatial predictions of the prediction error.

## 2.5  Implementation

The LPD approach as well as the simulation and case study were implemented using the R programming language version

4.x (R Core Team, 2023). Various packages were used including `FNN` (Beygelzimer et al., 2023) for distance calculations, `geodata` (Hijmans et al., 2023) for downloading the spatial data sets, `virtualspecies` (Leroy et al., 2016) for response generation in the simulation study, `caret` (Kuhn and Max, 2008), `randomForest` (Liaw and Wiener, 2002) and `ranger` (Wright and Ziegler, 2017) for the model training, `sf` (Pebesma, 2018) and `terra` (Hijmans, 2023) for the vector and raster data handling, and `plotly` (Sievert, 2020) and `ggplot2` (Wickham, 2016) for visualizations. The method to estimate the DI

and the AOA are implemented in the R package `CAST` (Meyer et al., 2024a). We also implemented the method to derive the LPD in the CAST package.

The supplementary simulation and case study are available at https://github.com/fab-scm/LPD.



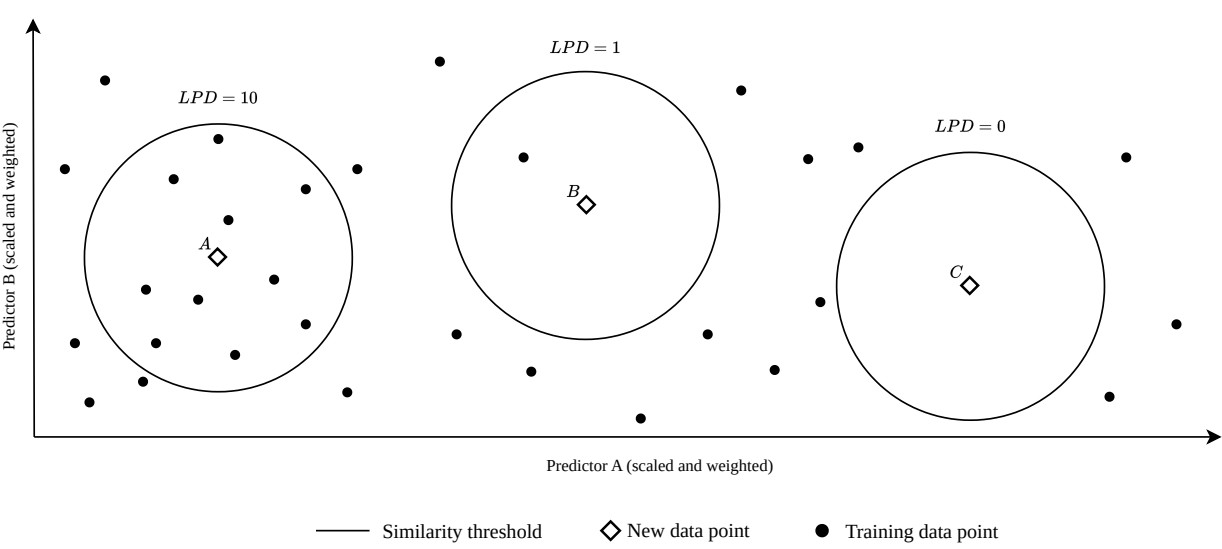

**Figure 2.** LPD visualization in a two-dimensional predictor space with three scenarios where a new data point (diamond) is supported by several training data points (A, LPD = 10), by only a single training data point (B, LPD = 1) or by no training data (C, LPD = 0) under consideration of the similarity threshold. The threshold remains constant for a specific model case, as it is dependent on the training data and the cross-validation strategy.





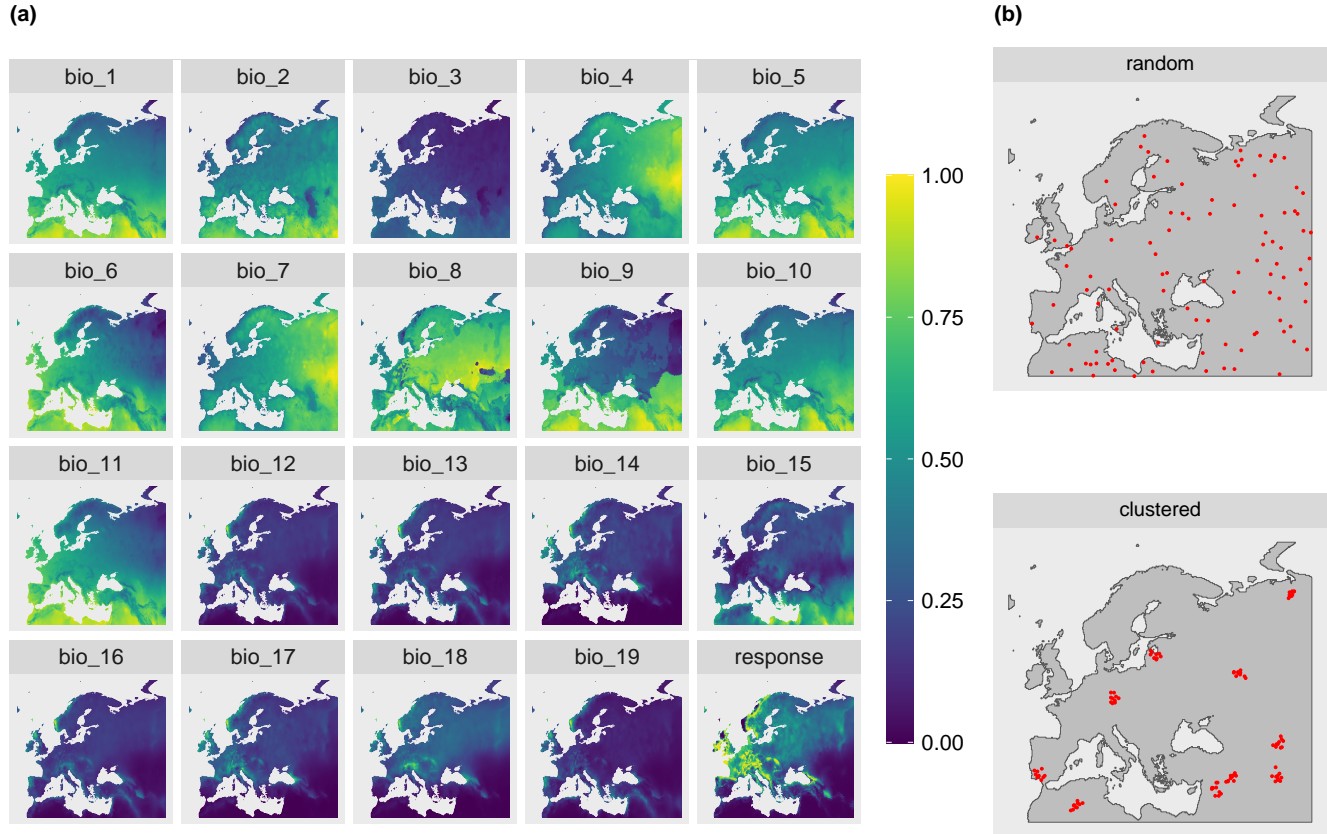

**Figure 3.** (a) The 19 predictor variables from the WorldClim dataset and the response variable used in the simulation study. All grids were stretched here from 0 to 1 for better visualization. (b) The two different sampling designs used in the simulation study (*random* and *clustered*).



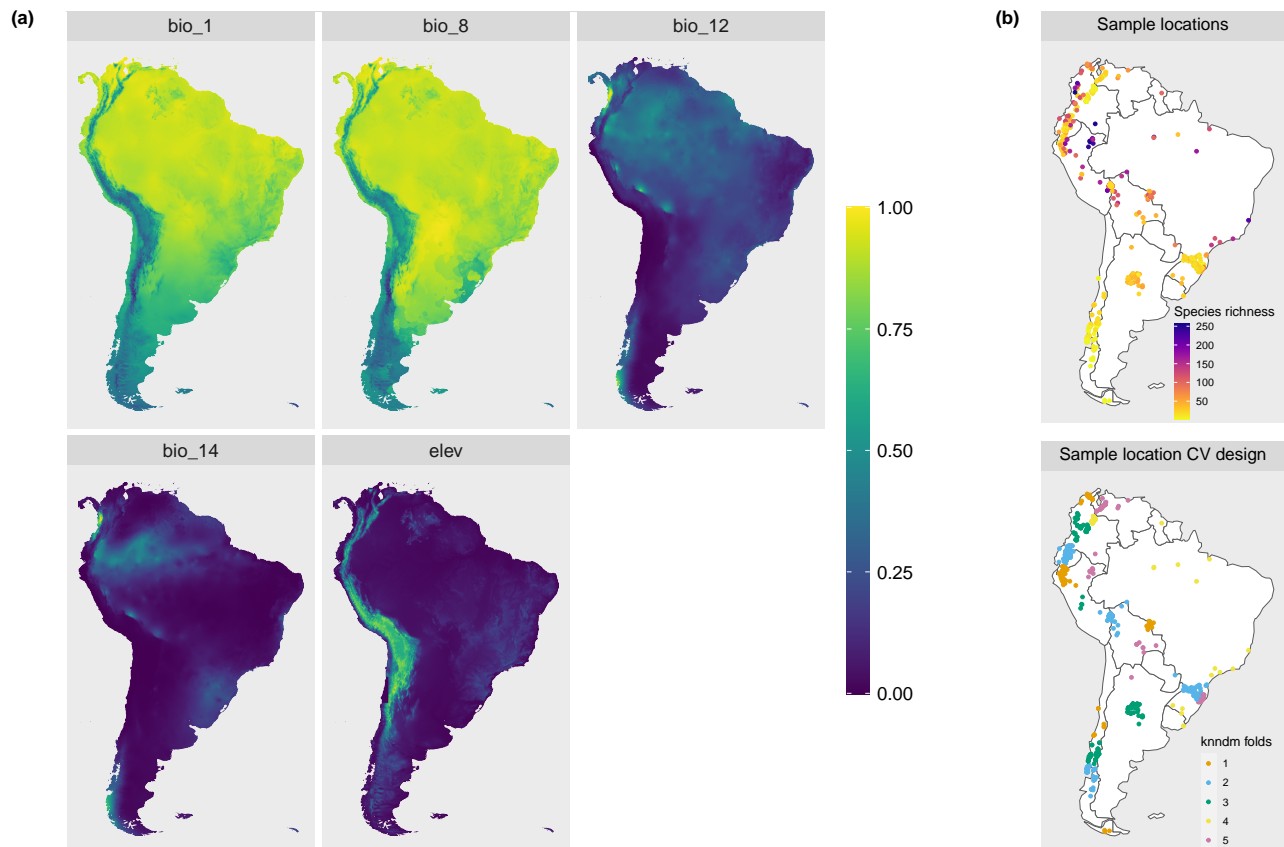

**Figure 4.** Data used in the case study. (a) The predictor variables selected by the spatial variable selection. (b) The training locations with species richness represented by color (top) and the cross-validation folds based on 5-fold Nearest Neighbor Distance Matching (bottom).



## 3   Results

### 3.1   Simulation study

Cross-validation indicated that the models have a high predictive performance, in both sampling scenarios. For the *random* sampling design, an $R^2$ of 0.96 and a RMSE of 0.05 was obtained via kNNDM cross-validation (which here equals random cross-validation). The *clustered* training data led to a cross-validated $R^2$ of 0.81 and an RMSE of 0.11. Since in the simulation study, the true values were known, the cross-validation estimates could be compared to the true error of the continuous predictions (see predictions in Figure A1) that was $R^2 = 0.83$, RMSE=0.10 using the model based on random samples and

$R^2 = 0.79$, RMSE=0.11 for the clustered design, respectively (see Table 1). When restricted to the AOA (LPD > 0), the true RMSE was 0.044 for the *random* and 0.078 for the *clustered* design (Table 1).

  The average LPD within the training data (revealed via cross-validation) was 15 for *random* and 26 for the *clustered* sampling scenario, with a standard deviation of 10 for both scenarios (see Table 1). Within the prediction data, the average LPD was 18 and the standard deviation was 12 for the random sampling the scenario. For the clustered sampling scenario the average LPD

was 31 with a standard deviation of 11.

  The AOA was largest for the *clusterd* sampling scenario with 96.1% of the prediction data points inside the AOA and slightly smaller for the *random* scenario with 90.1% inside the AOA (see Table 1 and Figure 8). The LPD (Figure 5) showed that the number of similar training data points was largest in the middle and north-eastern parts of the prediction area for the *random* scenario, whereas for the *clustered* scenario the highest coverage was reached in the southern parts of the prediction area.

With increasing LPD values, both the range of error values of the individual predictions and the average prediction error (RMSE, black points) decreased (see Figure 6). Areas within the AOA but with low LPD exceeded the cross-validation error while areas with high LPD significantly undercut the cross-validation error (see Figure 6).

  Strong relationships could be revealed between the DI and the cross-validation performance (RMSE) as well as between the LPD and the cross-validation performance (RMSE), which we both modelled using shape-constrained additive models

(Figure 7). The relationship between DI and the performance (RMSE) could be confirmed here with an $R^2$ of 0.57 and an RMSE of 0.013 between the predicted RMSE (blue line) and the true RMSE (see red points in first column of figure 7) for the *random* sampling scenario and an $R^2$ of 0.40 and an RMSE of 0.030 for the *clustered* scenario. However, the true RMSE was underestimated especially in areas with average DI values and overestimated in areas with higher DI values for the *random* scenario. For the *clustered* scenario, the true RMSE was underestimated for higher DI values. The second column in figure 7

shows the relationship between the LPD and the RMSE for the two scenarios. The agreement could be described here by an $R^2$ of 0.74 and an RMSE of 0.011 for the *random* and an $R^2$ of 0.71 and an RMSE of 0.020 for the *clustered* design. The uncertainty for higher DI values in the relationship with the performance (RMSE) could be explained by the LPD. Higher residuals (see third column in Figure 7) from the DI model were more commonly related to lower LPD values, i.e. to a lower coverage by the training data.

Continuous performance estimates were then derived by applying the models to DI and LPD for the entire prediction area (Figure 8). Comparing the spatial patterns for the predicted model performances (RMSE), similarities could be identified





between the DI-based and the LPD-based prediction of the *random* sampling scenario (see Figure 8a). Both predictions fur-
thermore showed a high visual agreement with the true prediction error. For the *clustered* scenario, the comparison between
the DI and LPD-based prediction showed greater differences. Especially in areas where high prediction errors were observed,

the LPD-based error prediction better reflected the true prediction error than the DI-based. Nevertheless, there are also areas
where the performance model based on the LPD highly overestimated the true prediction error (see Figure 8b).

## 3.2    Case study

The trained random forest model showed a moderate ability to predict species richness with an $R^2$ of 0.53 and an RMSE
of 31.97, obtained from kNNDM cross-validation. The model was based on 5 predictor variables, as selected during spatial

variable selection, with annual mean temperature being the most important predictor variable (see Figure B1). The average
LPD of the cross-validated training data was 40 (max: 126) with a standard deviation of 30.

     The DI values of the prediction area (Figure 9) ranged from 0 to 1.72 with an average of 0.13 and a standard deviation of
0.11. Highest DI values occurred in the Patagonia region, in parts of the Andes, and on smaller stretches of the west coast,
with some of these areas exceeding the AOA threshold (Figure 9c). The LPD values (Figure 9b) ranged from 0 to 163 with an

average value of 50 and a standard deviation of 29. The same regions that were obvious in the DI can also be recognized here,
as they are characterized by relatively low LPD values or even an LPD of 0 (i.e. outside the AOA). However, some areas had
low DI values but low LPD at the same time, hence are supported by few training data points only. This pattern is particularly
evident in the western region of the Amazon. Figure 10 shows the relationships between the DI and LPD of the cross-validated
training data and the corresponding random forest model performance (RMSE) using shape-constrained additive models. The

estimated performance, derived by applying the models to DI and LPD for the entire prediction area of South America is shown
in figure 11. Obvious deviations between spatial patterns of DI and LPD are especially found in the Amazon rainforest where,
based on the LPD, higher prediction errors are to be expected.

**Table 1.** Cross-validation and true prediction errors for the two scenarios of the simulation study.

| Sampling design | CV | | | Truth (overall) | | | Truth (inside AOA) | | |
|---|---|---|---|---|---|---|---|---|---|
| | $R^2$ | RMSE | MAE | $R^2$ | RMSE | MAE | $R^2$ | RMSE | MAE |
| random | 0.956 | 0.054 | 0.030 | 0.826 | 0.101 | 0.044 | 0.963 | 0.044 | 0.027 |
| clustered | 0.807 | 0.105 | 0.054 | 0.794 | 0.109 | 0.064 | 0.893 | 0.078 | 0.055 |





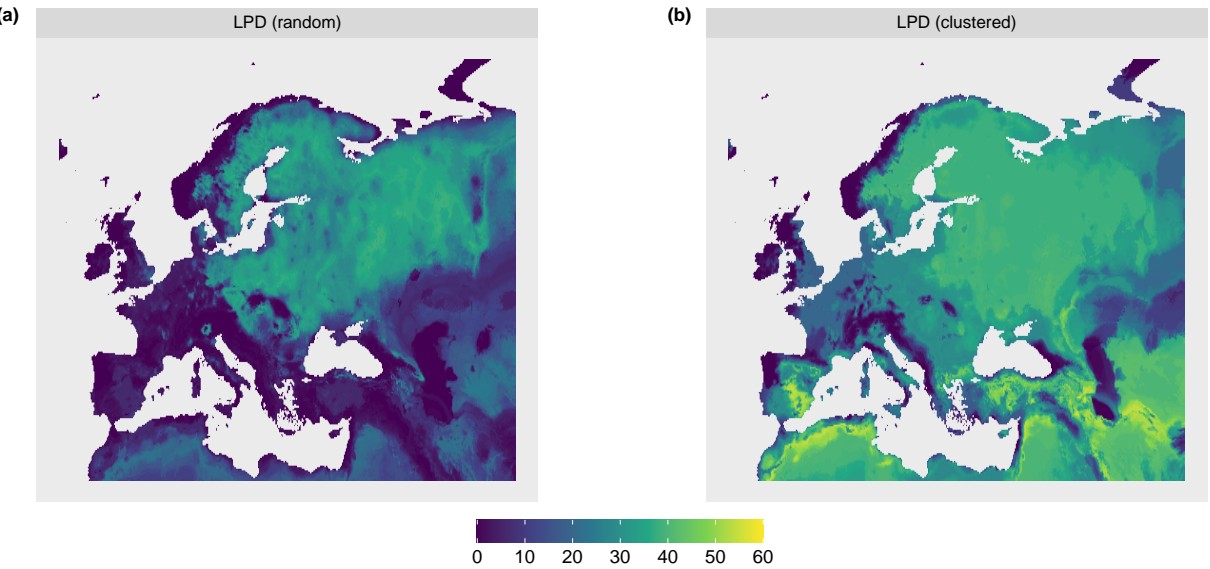

**Figure 5.** Local data point density (LPD) for the (a) random and (b) clustered.

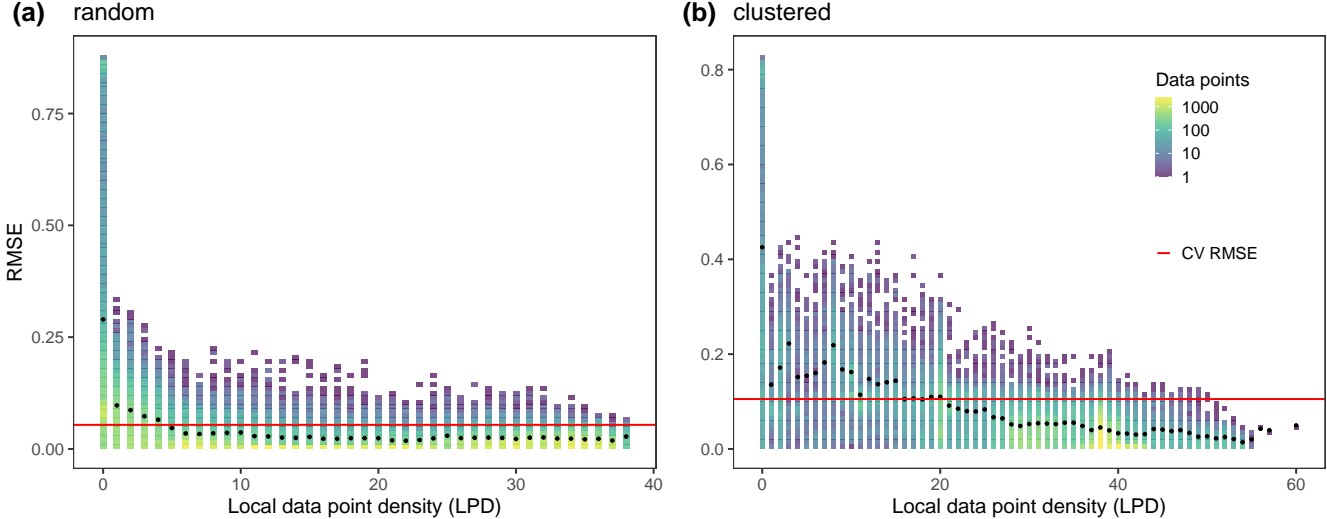

**Figure 6.** LPD values of the prediction locations (i.e. each pixel in the domain) for each of the two scenarios of the simulation study. LPD data bins are plotted against the respective true absolute error values; The color of a bin indicates the count of data it contains. The red line indicates the RMSE obtained from cross-validation in each scenario and the black points show the RMSE for all prediction locations with the respective LPD value. (a) is referred to the *random* and (b) to *clustered* sampling design.




**Table 2.** Similarity threshold and summary statistics of the LPD values for the two scenarios of the simulation study

| Sampling design | Similarity threshold | Training data (cross-validation) | | Prediction data | | % inside AOA |
| --- | --- | --- | --- | --- | --- | --- |
| | | avrgLPD $\pm \sigma$ | maxLPD | avrgLPD $\pm \sigma$ | maxLPD | |
| random | 0.411 | 15 ± 10 | 34 | 18 ± 12 | 38 | 90.1% |
| clustered | 0.771 | 26 ± 10 | 44 | 31 ± 11 | 60 | 96.7% |

**(a)** random

**(b)** clustered

**Figure 7.** Relationship between the error metric (RMSE) and the dissimilarity index (DI) (left), between the error metric (RMSE) and the local data point density (LPD) (middle) and between the absolute residuals of the DI model ($|$ truth $-$ model $|$) against the corresponding LPD values (right) both for the (a) random and (b) clustered sampling scenario; Only areas within the AOA are considered. Each single data point (black dot) corresponds to the RMSE from a sliding window of size 5, either along the DI axis or along the LPD axis. The shape constrained additive models are shown as a dashed blue line. The true RMSE which was calculated using the reference map and corresponding predictions within the identical windows of DI and LPD values is shown as red dots. The residuals of the DI model (right) are modelled as a general additive model (blue line) along the LPD axis.



**(a) random**

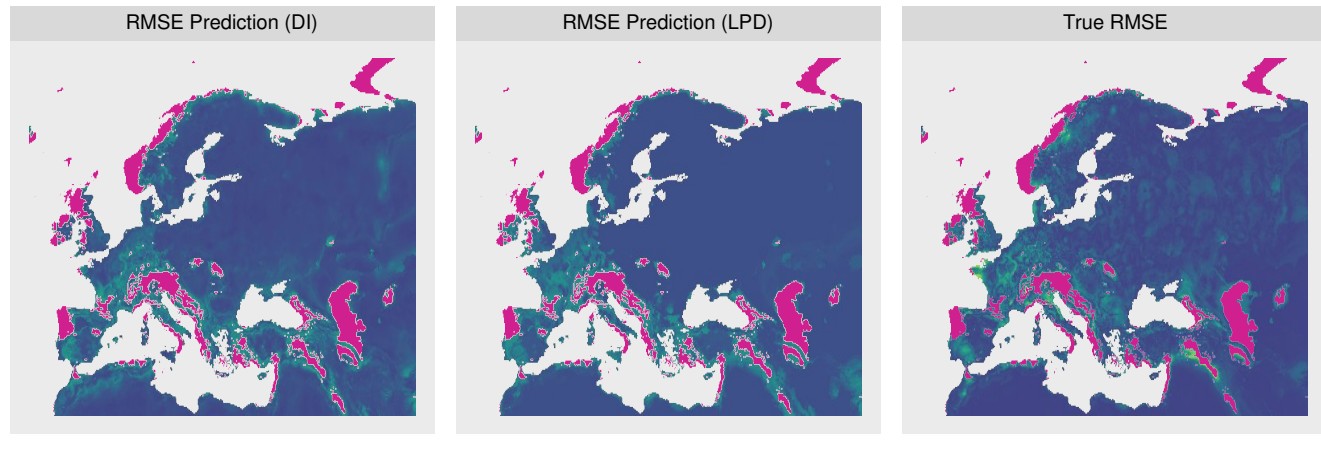

**(b) clustered**

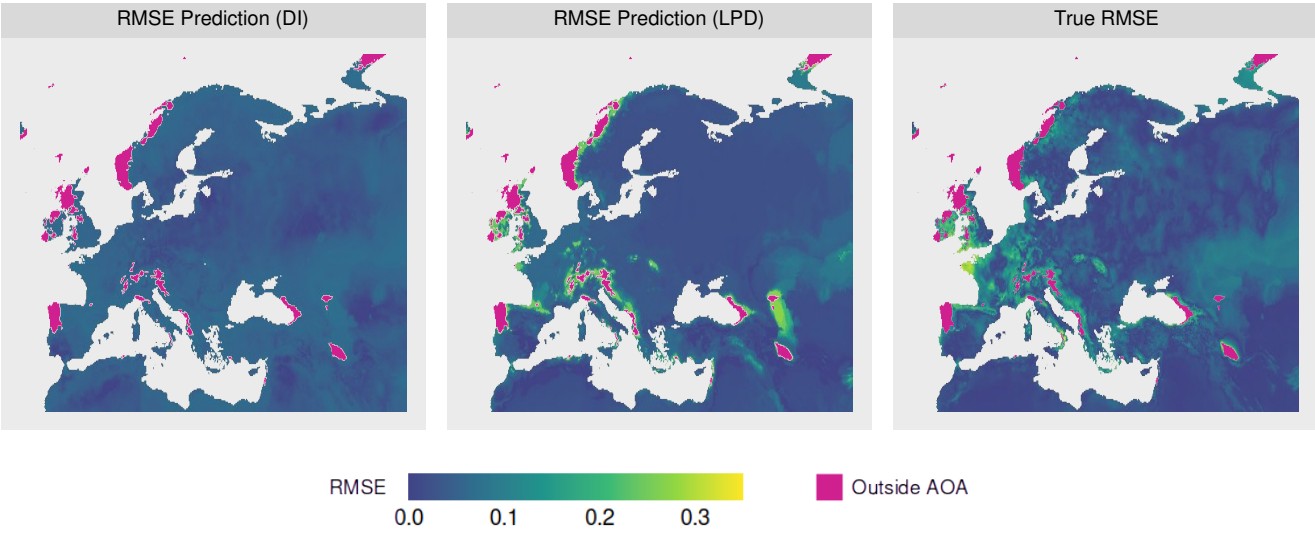

**Figure 8.** Model performance (RMSE) predictions based on the relationship of the model error to the DI (left) and LPD (middle) for the (a) *random* and (b) *clustered* scenario. The true prediction error (right) of the two scenarios is shown for comparison.



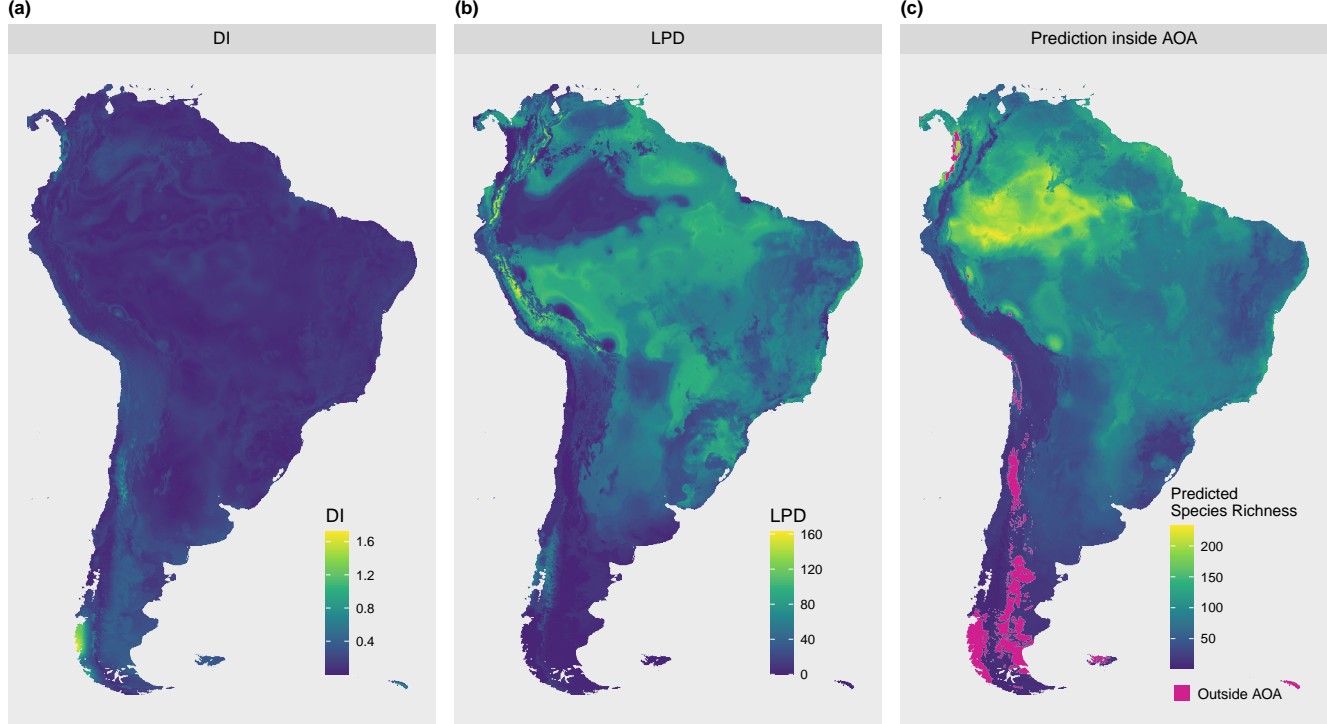

**Figure 9.** (a) DI, (b) LPD and (c) predictions of species richness inside the AOA.



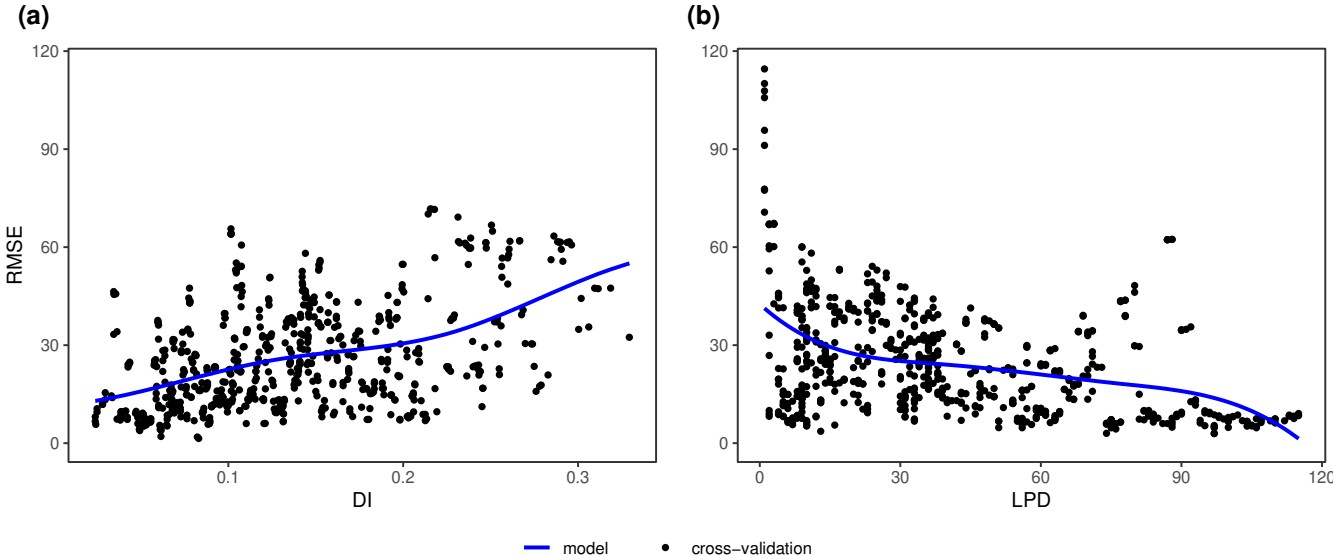

**Figure 10.** Relationship between the error metric (RMSE) and (a) the dissimilarity index (DI) and (b) the local data point density (LPD) for the case study; Only areas within the AOA are considered. Each data point corresponds to the RMSE from a sliding window of size 5, either along the DI axis, or along the LPD axis. The shape constrained additive models are shown as a blue line.

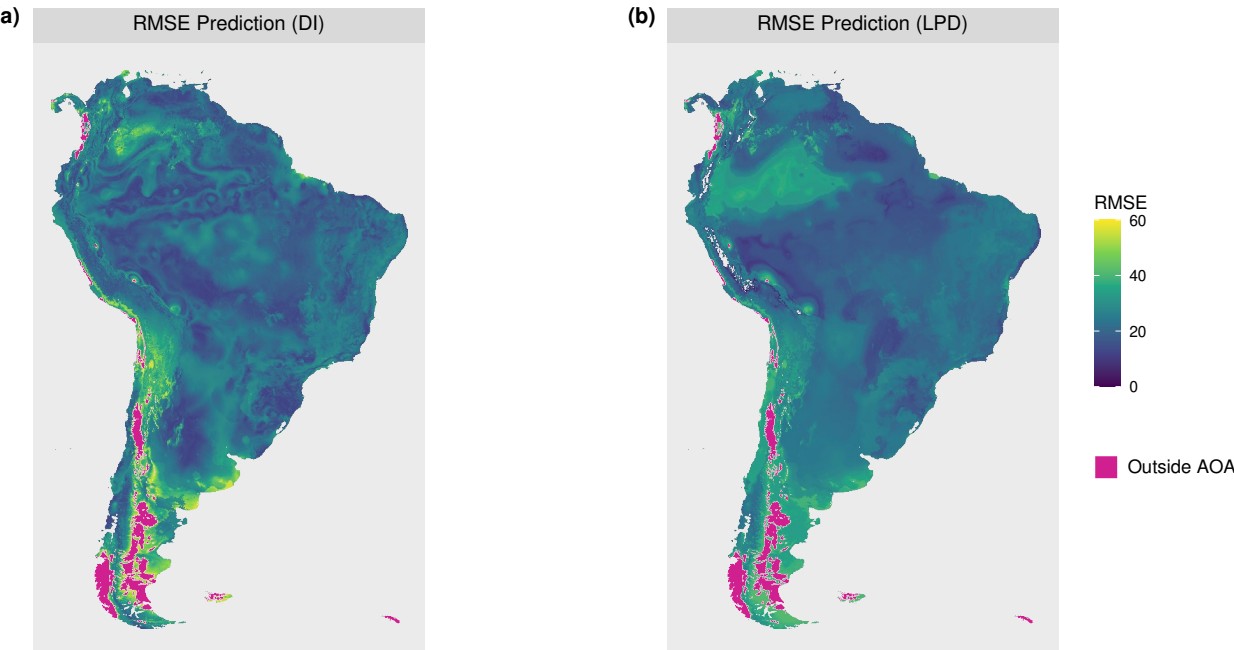

**Figure 11.** Predicted RMSE values inside the area of applicability (AOA) of the model. Predictions are based on (a) DI and (b) LPD. Model performance (RMSE) predictions based on the relationship of the model error to the (a) DI and (b) LPD.



## 4 Discussion and conclusions

We suggested a method to calculate a local data point density measure to support the assessment of models applicability and
the reliability of their spatial predictions. The method addresses the limitation of the previously suggested nearest neighbor
distance measures, such as the DI of Meyer and Pebesma (2021), where a single isolated training data point is treated the same
as an area in the predictor space that is densely covered by training data. The results of the simulation study confirmed that areas
with a low DI may still have significantly higher prediction errors which could be explained by differences in the LPD which
we identified as a stronger predictor for prediction performance, than the DI alone. The results highlight that while the cross-
validation performance applies, on average, within the AOA, there are significant variations based on the density of training
data coverage. Specifically, areas with high training data coverage (high LPD) exhibit significantly lower prediction errors,
indicating that the model was better trained for these regions. In contrast, areas with low LPD showed prediction performances
that were significantly lower than the indicated CV performance, suggesting that the model struggles to generalize well in
regions with sparse training data. Hence, we have shown that the LPD serves as a good indicator for prediction performance
and compensates for the limitations of the DI that is highly sensitive to outliers. Based on the results, users can be informed
about the expected reliability of predictions across the area of interest. Analogous to the relationship between DI and prediction
error, the modelled relationship based on the LPD may be used to restrict predictions to areas with a certain user-defined
performance.

To evaluate the suitability of the LPD to support uncertainty analysis of spatial predictions, we used simulated response
variables in the simulation study. Both simulation scenarios (*random* and *clustered*) resulted in predictions with high accuracy,
as the simulated response was a clear function of the predictor variables. Consequently, observed prediction errors could largely
be attributed to gaps in the coverage of environmental predictors. In scenarios with weaker dependencies, the relationship
between LPD and the prediction error will likely be less pronounced, as it could be observed in our simulation study, estimated
here from the cross-validated training data. This is because the primary source of uncertainty in these scenarios is often not
a lack of coverage in the predictor space but rather missing knowledge or missing availability of relevant predictors to model
the response. It is important to note that we do not expect that a low local training data point density necessarily leads to poor
prediction results, nor does a high LPD automatically guarantees high prediction performance. This largely depends on the
strengths or ambiguity of relationships in the respective part of the predictor space. The presence of a single training data point
may be sufficient to make predictions about the response if noise is minimal and this point is located in an area of the predictor
space where small environmental variations (reflected by the predictors) do not lead to significant changes in the response.
However, considering the LPD allows for estimating the probability of achieving high prediction performance.

The LPD can also aid in refining models by identifying areas with insufficient training data coverage, which can then be
addressed in future field sampling campaigns. Our case study demonstrates the advantage of the LPD in addition to the DI. For
example, in the Amazon rainforest, while the DI is low, the LPD clearly indicates that the predictions are based on very few
similar training data points, suggesting potentially insufficient coverage (see Figure 9).



While the LPD method shows promise in enhancing the assessment of prediction model, several aspects merit further exploration and consideration for future development:

1. Alternative similarity thresholds: the LPD highly depends on the similarity threshold. Here we used the threshold suggested to derive the area of applicability of a model. This threshold is dependent on the cross-validation strategy being used, since the similarity threshold is defined as the outlier-removed maximum DI observed during cross-validation. The CV strategy should hence be designed in line with the sampling distribution over the prediction area. For example, if a random cross-validation is used on spatially clustered data, the similarity threshold is likely to be rather small and the cross-validation performance is high because we are testing the ability of the model to make predictions within the clusters. As a consequence, large parts of the prediction area will fall outside of the AOA because the performance estimate only applies to a limited area. Since the same threshold is used to define similarity, these parts of the prediction area will have an LPD of 0 (outside the AOA). We therefore recommend using NNDM (Milà et al., 2022) or it's k-fold variant (Linnenbrink et al., 2024) to test the ability of the models to achieve the prediction task and, as a consequence, derive a suitable threshold defining similarity within the data set. Other concepts for deriving the similarity threshold values were not analyzed in this study but remain options for future investigation.

2. Include the LPD in the delineation of the AOA: we implemented the LPD to be available as additional area-wide information, but the method is not involved in the delineation of the AOA yet. Despite the continuously available information on DI and LPD, we still see the benefit of the binary AOA to limit predictions - to clearly indicate the applicability if the model. It might be a consideration to replace the DI in the delineation of the AOA by the LPD.

3. Deriving prediction uncertainty from the LPD, or DI and LPD: the relationship between LPD and the performance measure was elaborated in the simulation study (see Figure 7). Though other factors influence the prediction performance as well, and hence, we do not expect a perfect fit, we are confident that the LPD provides a valuable predictor of prediction uncertainties. In addition to the single relationship further investigations on a combined relationship of DI and LPD values and possibly other factors should be studied.

4. Scaling of the algorithm: for large training or prediction data sets, the algorithm needs very long computation times as distances in the multivariate feature space between all prediction and training data points are calculated. Limitation to a random subset of the training data or limiting the LPD to a user defined maximum may be considered to reduce computation times.

5. Full density calculation: the LPD represents a measure that indicates the local density by counting neighbours in a predefined area around a new data point in the multidimensional predictor data space. Therefore the LPD does not express a full density for the entire predictor space.

In summary, we proposed an approach to calculate area-wide training data point density in the predictor space - the local training data point densities (LPD). The method is implemented in the R package CAST (Meyer et al., 2024b). We suggest



communicating the LPD alongside spatial predictions or constraining predictions according to the LPD to increase the reliability of spatial predictions, especially for large-scale applications such as global predictions.

*Code and data availability.* The method to calculate the LPD inside the AOA methodology has been implemented and published in the R package CAST (Meyer et al., 2024b) which is available on CRAN. The simulation studies and all figures of this paper can be reproduced using the R-scripts available from https://github.com/fab-scm/LPD.

*Author contributions.* F.S., C.K. and H.M conceived the ideas with contributions of M.L.; F.S. implemented the methods and conducted the study. F.S. and H.M. wrote the manuscript with contributions of C.K. and M.L.

*Competing interests.* No competing interest were present in the process of this paper.

*Acknowledgements.* This study was partly funded by the DFG project Carbon4D (project number 455085607).



# Appendix A: Simulation Study

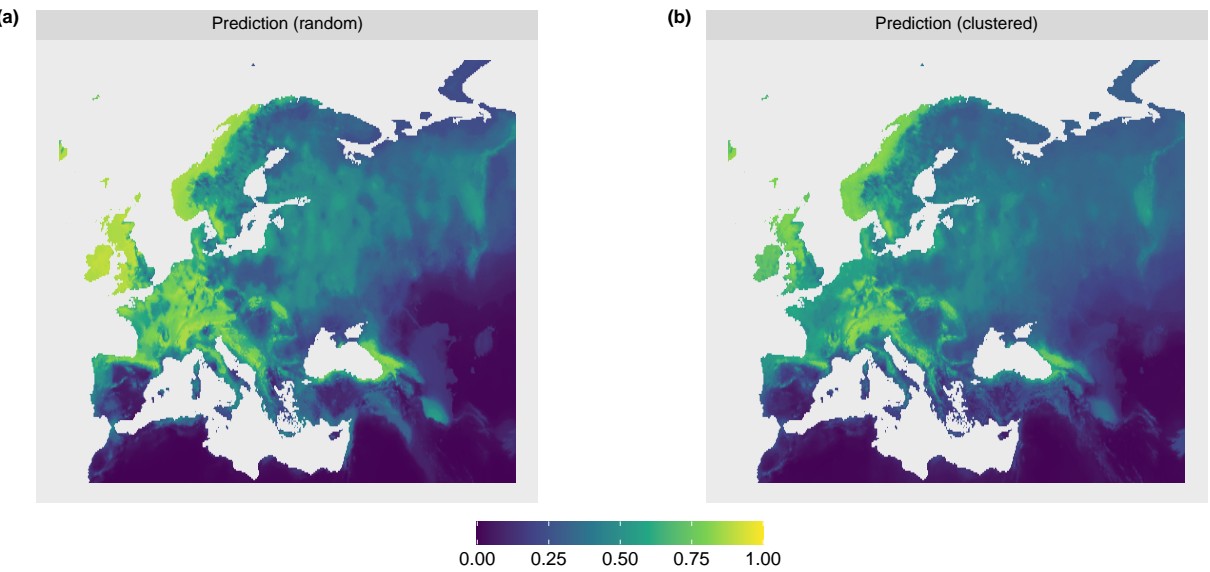

**Figure A1.** Predictions for the (a) random and (b) clustered sampling design.

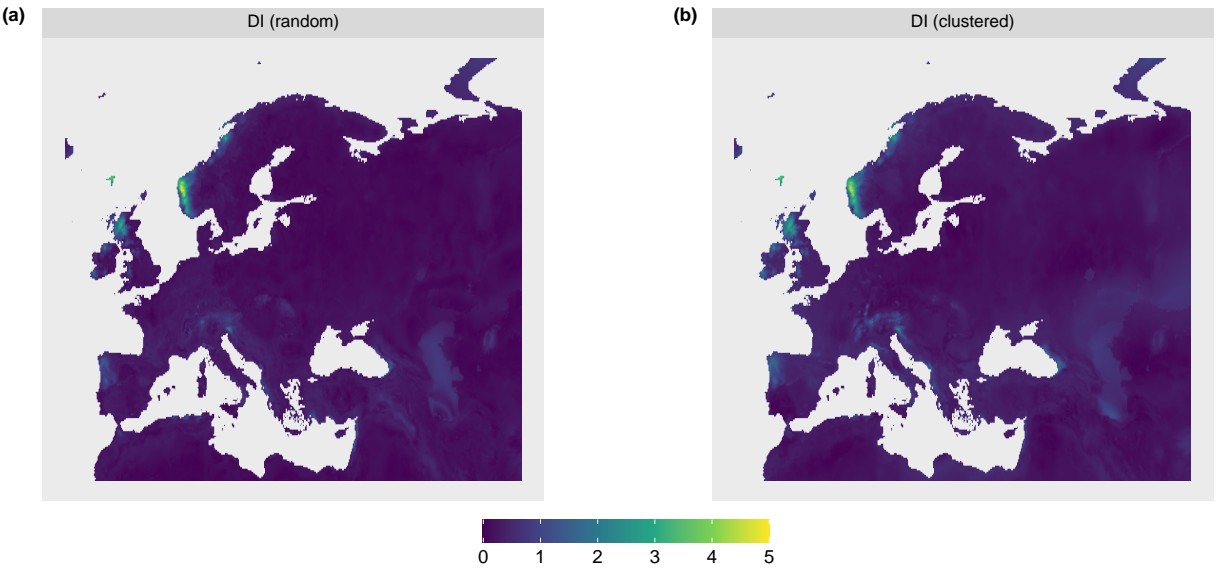

**Figure A2.** Dissimilarity index (DI) for the (a) random and (b) clustered sampling design.





**(a)** **(b)**

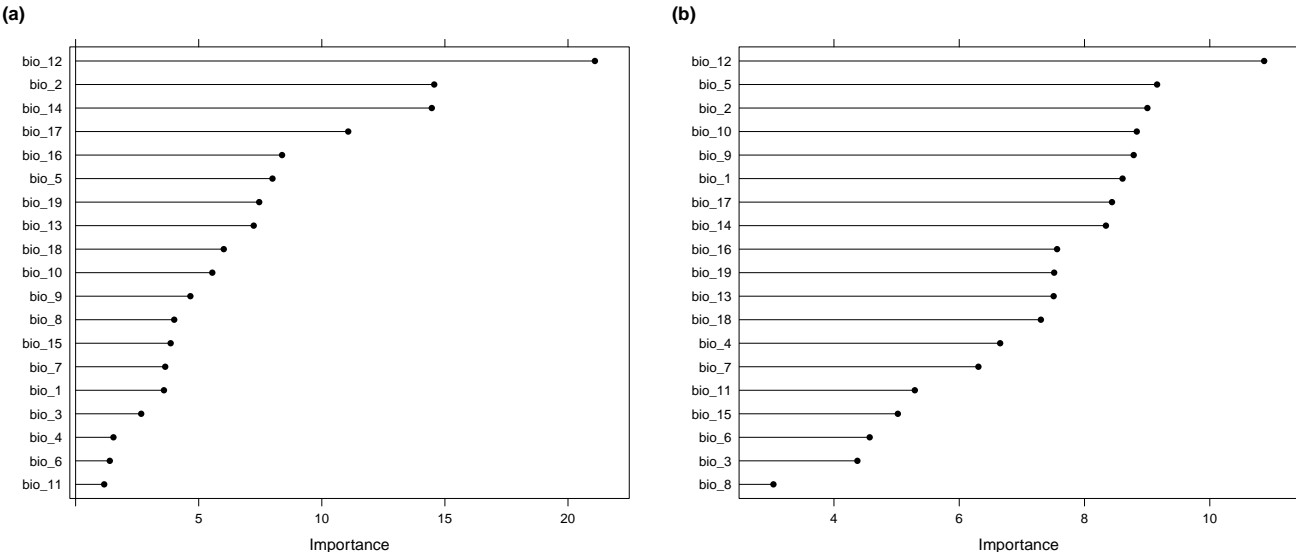

**Figure A3.** Variable importance of the *RF* model for the (a) random and (b) clustered sampling scenario.

## Appendix B:  Case Study

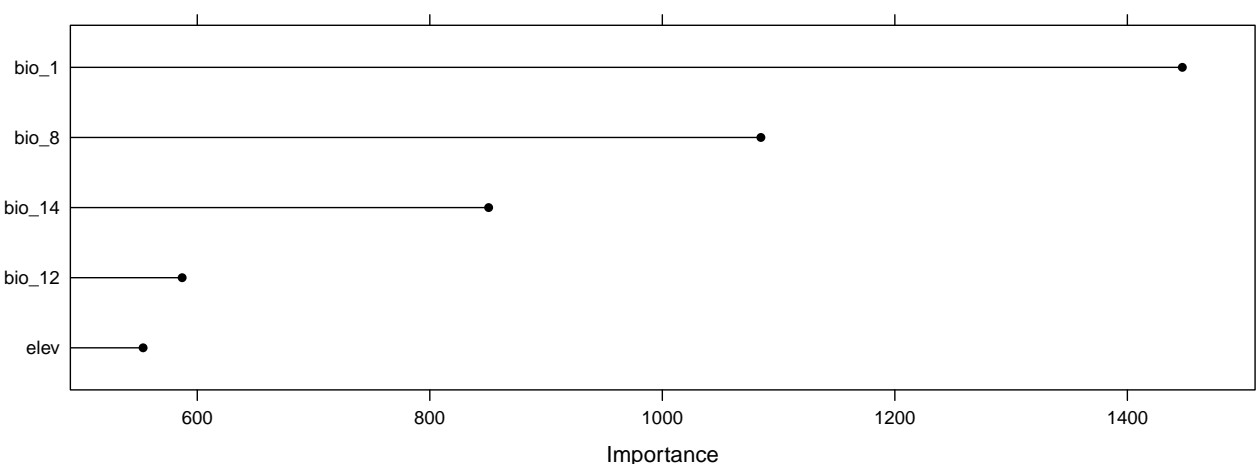

**Figure B1.** The estimated variable importance from the model training.



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
