# Peer review of "Estimation of local training data point densities to support the assessment of spatial prediction uncertainty"

_EGUsphere, 2024_

## Author Response (AR1)

**Author's Response**

**CEC1**

We have now archived the contents of the GitHub repository referenced in our paper, which contains all scripts and data, as well as the repository containing the R package CAST on Zenodo. In the revised version of our manuscript, we updated the "Code and Data Availability" section to ensure full compliance with the journal's data policy. Specifically, we provide links to permanent repositories containing all relevant code and data used in our study. The updated section looks like this:

*The current version of the method to calculate the introduced Local Point Density (LPD) is available from the developer version of the R package CAST (https://github.com/HannaMeyer/CAST) under the GNU General Public Licence (GPL >= v2). The exact R package version of the implementation used to produce the results of this paper is CAST Version 1.0.2 which is  published on CRAN (Meyer et al., 2024b) and Zenodo (https://doi.org/10.5281/zenodo.14362793). The exact version of the simulation study and the case study including all scripts as well as the input data used to run the models and produce the results and plots described in this paper is archived on Zenodo (https://doi.org/10.5281/zenodo.14356807) under the GNU General Public Licence  (GPL v3).*

In the current manuscript, CAST version 1.0.0 is cited. Since version 1.0.2, which addresses only minor issues, has now been archived on Zenodo, the revised versions of the manuscript reference this updated version.

**RC1**

***Major Scientific Concerns***

**1. Applicability of LPD to non-fixed sampling locations and smaller-scale cases**

We acknowledge the reviewer's concern regarding the applicability of the LPD approach beyond large-scale spatial interpolation with fixed sampling locations. While the case studies in the manuscript focus on large-scale applications, the LPD method is not inherently restricted to such contexts. It is based on the local density of training points in the predictor space rather than geographic space, making it applicable across sampling strategies, including non-fixed and smaller-scale settings. Further explanation has been added to the discussion of the revised manuscript (**ll. 278-283**).

We demonstrated its applicability to local studies in the help file of the CAST package ( `?` `AOA` ). In this example, we use soil moisture and temperature logger data at the farm scale to show how the LPD can identify areas that are well or poorly covered by training data (in the predictor space).

**2. How does RF incorporate LPD and influence tree structure?**

The reviewer raises an important question about how Random Forest (RF) integrates spatial patterns and whether this necessarily leads to model improvement. While RF does not inherently encode spatial information—unless explicitly incorporated (see https://doi.org/10.5194/gmd-17-6007-2024)—it captures relationships within the predictor variables, which can include local density information.

The LPD does not influence the model itself and is not intended to be incorporated into the training process. Instead, it serves solely as a measure of training point density in the predictor data space for a new prediction location. Since it is not used during model training, it does not alter the tree structure.

Our goal is therefore not to improve model accuracy directly, but rather to enhance the reliability of predictions by identifying areas with high training coverage or by quantifying potential prediction accuracy loss related to low LPD.

We have added further clarification in the abstract (**ll. 16-18**) and the discussion (**ll. 294-297**) of the revised manuscript, emphasizing that while the LPD does not modify the tree structure (as it is not included in training) it offers a complementary measure of uncertainty.

**3. Comparison with traditional spatial interpolation methods like Kriging**

We appreciate the suggestion to better integrate our work within the broader context of spatial interpolation methods. However, our focus is on machine learning–based approaches, where, unlike kriging, which uses the kriging variance to quantify spatial uncertainty based on spatial autocorrelation, there is no established method for uncertainty assessment. We believe that comparing the LPD to uncertainty assessment techniques used in interpolation methods is not particularly helpful, as the models rely on different assumptions and strategies. We prefer to maintain our focus on machine learning methods in this context.

Nevertheless, we see the benefit of working out the differences between the various approaches and have elaborated on this in the discussion of the revised manuscript (**ll. 297-310**).

**4. Identifying scenarios where LPD is most effective and areas for improvement**

We fully agree that no single method is universally optimal. In our manuscript we also compare it to the dissimilarity index that was proposed earlier. To provide a clearer guideline for practitioners, we have added information to the manuscript discussing the conditions under which LPD performs best (**ll. 288-293**) . These include:

- Cases with heterogeneous predictor distributions where training data density varies significantly.
- Applications where it is crucial to assess model applicability beyond training locations.
- Datasets with a large number of predictors in which traditional statistical density measures, such as kernel density estimation, fail to capture data density effectively due to high computational costs and inefficiency in high-dimensional spaces.

**5. Missing summary table for 19 bio variables**

We acknowledge this oversight and have included a summary table detailing all 19 bio variables used in our study. This table provides variable names and descriptions and is placed in the appendix (**p. 23**).

**RC2**

No Major concerns were mentioned by this reviewer.

**RC3**

**1. *"The study focuses on random forest models... wondering if LPD can perform well with other algorithms."***

We agree that the behavior of different machine learning algorithms with respect to extrapolation and sensitivity to training data density can vary. To demonstrate that our LPD approach is not limited to random forest models, we have included an example using Support Vector Machines in the helpfile of the AOA function in our CAST package.

The results show that while the overall LPD patterns remain similar across algorithms, minor differences can emerge. These are primarily due to the varying ways in which different models internally weight predictor variables. Such differences are expected, as variable importance influences both model predictions and the LPD outcome.

When we neutralize these differences - by assigning equal weights to all variables and applying the same cross-validation strategy across models - the resulting LPD patterns become identical. This is because the LPD results are driven by the location of the training data and new data in the predictor pace well as  the cross-validation design, rather than by the specific characteristics of the learning algorithm (see supplement figures).

We have added a brief discussion (**ll. 268-277**) of this point to the revised manuscript to clarify that LPD can be used with different algorithms as it is independent of the model's

internal mechanics and mainly the variable importances as well as the cross-validation folds are used for the LPD calculation - alongside the predictor values of the training data and the new data.

**2. "Real-world geoscience data can show unphysical teleconnections... Will this pose a concern for the application of the LPD method?"**

We use the predictor variables selected by the model - weighted by their estimated relevance - without distinguishing between causal and non-causal relationships, because the model itself does not make that distinction either. If a non-causal (i.e., potentially unphysical) variable is relevant for the model's prediction, then being close in predictor space to the training data (i.e., having a high LPD) becomes even more important - because the model is extrapolating based on a spurious relationship, which is likely to fail outside known data. Such failures are typically detectable through spatial cross-validation strategies, which we recommend using to define the similarity threshold and to assess model performance under realistic prediction scenarios. We have added further explaination in the discussion of our revised manuscript (**ll. 311-316**)

**3. "The simulation study... would be more convincing if LPD were tested on a wider range of simulated cases like non-Gaussian or multimodal responses."**

Thank you for pointing this out. We agree that further testing with more complex or non-Gaussian response structures would enhance the validation. We will acknowledge in the discussion that evaluating LPD under, e.g. multimodal or skewed relationships is essential for broader generalizability. We have added this as a potential extension of our work and included it under "future development" in our discussion (**ll. 340-346**)

**4. "Some technical terms (e.g., shape-constrained additive model) may be unfamiliar..."**

We added brief explanations at the mention of shape-constrained additive models (**Section 2.3, ll. 150-155**) to ensure accessibility to a broader readership.

**Grammar and clarity improvements**

We have again carefully reviewed the manuscript for grammatical and clarity improvements and made smaller adjustment.

**Legend:** Chief Editor Comment #1 Referee Comment #1 Referee Comment #3

[revised manuscript text omitted]

Although this study focused on random forest models due to their widespread use in spatial prediction, the LPD approach is not restricted to this algorithm. Since LPD is computed post hoc based on predictor space distances and does not rely on the models internal structure, it can be applied to other machine learning models such as e.g. gradient boosting or support vector machines. Thus, LPD offers a model-agnostic measure of training data density in predictor space. We included an example using support vector machines in the help file of the CAST package to demonstrate this flexibility. Note that under controlled conditions (e.g., uniform weights and consistent cross-validation design), the LPD is identical across models. However, variable

importance weighting depends on the model being used and the similarity threshold is based on the cross-validation strategy. Hence, for an identical set of training data, the LPD may vary across models. This is expected and reasonable, as differences in variable weighting influence model predictions and should therefore also influence the LPD outcomes when more weight is placed on certain variables.

While our case studies focus on large-scale applications with fixed training data distributions, the LPD method is not limited to such settings. Its core principle  quantifying training data density in the predictor space  is independent of the geographic scale or spatial arrangement of the sampling locations. In fact, LPD can be particularly useful in small-scale studies or adaptive sampling designs, where training data locations are not fixed and may evolve based on field conditions. A practical example of such a smaller-scale application is included in the help file of the CAST package, which demonstrates the use of LPD with soil moisture and temperature logger data at the farm scale.

The LPD can also aid in refining models by identifying areas with insufficient training data coverage, which can then be addressed in future field sampling campaigns. Our case study demonstrates the advantage of the LPD in addition to the DI. For example, in the Amazon rainforest, while the DI is low, the LPD clearly indicates that the predictions are based on very few similar training data points, suggesting potentially insufficient coverage (see Figure 9).

Therefore LPD primarily offers added value for:

1. Cases with heterogeneous predictor distributions where training data density varies significantly.

2. Applications where it is crucial to assess model applicability beyond training locations.

3. Datasets with a large number of predictors in which traditional statistical density measures, such as kernel density estimation, fail to capture data density effectively due to high computational costs and inefficiency in high-dimensional spaces.

It is important to note that the LPD is not used as an input variable and does not influence the training process of the machine learning model itself. The LPD is calculated after model prediction to quantify, for each prediction point, how densely the predictor space is covered by training data. It provides an additional, model-independent measure to assess where predictions are likely more or less reliable based on the available training data coverage. It can be used similarly to uncertainty measures in geostatistical interpolation methods, such as Kriging, which provides native estimates of prediction uncertainty through the Kriging-variance. The Kriging variance at a given location depends solely on the model of spatial autocorrelation and the location of sample points and is completely independent of the observed values at those points. As such, it is widely used to evaluate how sample spacing, density, or orientation affect the precision of spatial estimates, and plays a crucial role in guiding sampling and monitoring designs. In contrast, machine learning models do not typically make assumptions about spatial autocorrelation, nor do they provide a native uncertainty metric analogous to the Kriging variance. While recent developments such as the DI and AOA help assess model extrapolation risk in predictor space, they consider only the distance to the nearest training point. The LPD builds upon this by quantifying how many similar training points are present within a

defined similarity threshold in the predictor space. Although arising from different modeling paradigms (statistical modeling vs. machine learning), both Kriging variance and the LPD serve a similar conceptual function as they highlight areas of greater or lesser confidence in prediction based on data support. In this way, the LPD complements machine learning workflows by supporting the assessment of prediction reliability and can likewise be used to inform sampling strategies or to mask uncertain

310    regions in spatial prediction maps.

We acknowledge that in geoscientific applications, models may incorporate predictor variables that show spurious correlations that arise due to data artifacts or oversampling. Since LPD uses the predictors selected and weighted by the model, it inherits any such issues. However, this makes LPD particularly valuable: if a model's prediction relies heavily on such non-causal predictors, then high local training data density becomes even more important to reduce the risk of poor extrapolation.

315    The use of spatial cross-validation, which we recommend for defining the AOA threshold, can help to identify overfitting to such spurious structures.

While the LPD method shows promise in enhancing the assessment of prediction models, several aspects merit further exploration and consideration for future development:

1. Alternative similarity thresholds: the LPD highly depends on the similarity threshold. Here we used the threshold sug-

320    gested to derive the area of applicability of a model. This threshold is dependent on the cross-validation strategy being used, since the similarity threshold is defined as the outlier-removed maximum DI observed during cross-validation. The CV strategy should hence be designed in line with the sampling distribution over the prediction area. For example, if a random cross-validation is used on spatially clustered data, the similarity threshold is likely to be rather small and the cross-validation performance is high because we are testing the ability of the model to make predictions within the clus-

325    ters. As a consequence, large parts of the prediction area will fall outside of the AOA because the performance estimate only applies to a limited area. Since the same threshold is used to define similarity, these parts of the prediction area will have an LPD of 0 (outside the AOA). We therefore recommend using NNDM (Milà et al., 2022) or it's k-fold variant (Linnenbrink et al., 2024) to test the ability of the models to achieve the prediction task and, as a consequence, derive a suitable threshold defining similarity within the data set. Other concepts for deriving the similarity threshold values were

330    not analyzed in this study but remain options for future investigation.

2. Include the LPD in the delineation of the AOA: we implemented the LPD to be available as additional area-wide information, but the method is not involved in the delineation of the AOA yet. Despite the continuously available information on DI and LPD, we still see the benefit of the binary AOA to limit predictions - to clearly indicate the applicability if the model. It might be a consideration to replace the DI in the delineation of the AOA by the LPD.

335    3. Deriving prediction uncertainty from the LPD, or DI and LPD: the relationship between LPD and the performance measure was elaborated in the simulation study (see Figure 7). Though other factors influence the prediction performance as well, and hence, we do not expect a perfect fit, we are confident that the LPD provides a valuable predictor of prediction uncertainties. In addition to the single relationship further investigations on a combined relationship of DI and LPD values and possibly other factors should be studied.

4. Simulation study with non-Gaussian or multimodal responses: In our simulation we used the `virtualspecies` package to generate a synthetic response variable by computing a principal component analysis (PCA) on environmental predictor variables, followed by applying Gaussian functions to the first two PCA axes. We acknowledge that further testing with more complex response structures such as non-Gaussian, multimodal, or skewed relationships would enhance the validation and help generalize the applicability of LPD. We therefore propose this as a valuable extension for future work, particularly to better understand the performance of LPD under more heterogeneous or ambiguous ecological responses.

5. Scaling of the algorithm: for large training or prediction data sets, the algorithm needs very long computation times as distances in the multivariate feature space between all prediction and training data points are calculated. Limitation to a random subset of the training data or limiting the LPD to a user defined maximum may be considered to reduce computation times.

6. Full density calculation: the LPD represents a measure that indicates the local density by counting neighbours in a predefined area around a new data point in the multidimensional predictor data space. Therefore the LPD does not express a full density for the entire predictor space.

In summary, we proposed an approach to calculate area-wide training data point density in the predictor space - the local training data point densities (LPD). The method is implemented in the R package CAST (Meyer et al., 2024b). We suggest communicating the LPD alongside spatial predictions or constraining predictions according to the LPD to increase the reliability of spatial predictions, especially for large-scale applications such as global predictions.

*Code and data availability.* The current version of the method to calculate the introduced Local Point Density (LPD) is available from the developer version of the R package CAST (https://github.com/HannaMeyer/CAST) under the GNU General Public Licence (GPL >= v2). The exact R package version of the implementation used to produce the results of this paper is CAST Version 1.0.2 which is published on CRAN (Meyer et al., 2024b) and Zenodo (https://doi.org/10.5281/zenodo.14362793). The exact version of the simulation study and the case study including all scripts as well as the input data used to run the models and produce the results and plots described in this paper is archived on Zenodo (https://doi.org/10.5281/zenodo.14356807) under the GNU General Public Licence (GPL v3).

*Author contributions.* F.S., C.K. and H.M conceived the ideas with contributions of M.L.; F.S. implemented the methods and conducted the study. F.S. and H.M. wrote the manuscript with contributions of C.K. and M.L.

*Competing interests.* No competing interest were present in the process of this paper.

*Acknowledgements.* This study was partially funded in the course of the project Carbon4D (455085607) and the TRR 391 Spatio-temporal Statistics for the Transition of Energy and Transport (520388526), both by the Deutsche Forschungsgemeinschaft (DFG, German Research Foundation).

**Table A1.** Bioclimatic variables from the WorldClim dataset.

[revised manuscript text omitted]